# Conserved *miR396b-GRF* Regulation Is Involved in Abiotic Stress Responses in Pitaya (*Hylocereus polyrhizus*)

**DOI:** 10.3390/ijms20102501

**Published:** 2019-05-21

**Authors:** A-Li Li, Zhuang Wen, Kun Yang, Xiao-Peng Wen

**Affiliations:** The Key Laboratory of Plant Resources Conservation and Germplasm Innovation in Mountainous Region (Ministry of Education), Institute of Agro-Bioengineering and College of Life Sciences, Guizhou University, Guiyang 550025, China; alihg0519@163.com (A.-L.L.); gzu_zwen@163.com (Z.W.); kyanggz@163.com (K.Y.)

**Keywords:** pitaya, *miR396b*, phylogenetic analyses, *HpGRF6*, abiotic stress

## Abstract

*MicroRNA396* (*miR396*) is a conserved microRNA family that targets growth-regulating factors (GRFs), which play significant roles in plant growth and stress responses. Available evidence justifies the idea that *miR396*-targeted GRFs have important functions in many plant species; however, no genome-wide analysis of the pitaya (*Hylocereus polyrhizus*) *miR396* gene has yet been reported. Further, its biological functions remain elusive. To uncover the regulatory roles of *miR396* and its targets, the hairpin sequence of pitaya *miR396b* and the open reading frame (ORF) of its target, *HpGRF6*, were isolated from pitaya. Phylogenetic analysis showed that the precursor *miR396b* (*MIR396b*) gene of plants might be clustered into three major groups, and, generally, a more recent evolutionary relationship in the intra-family has been demonstrated. The sequence analysis indicated that the binding site of *hpo-miR396b* in *HpGRF6* is located at the conserved motif which codes the conserved “RSRKPVE” amino acid in the Trp–Arg–Cys (WRC) region. In addition, degradome sequencing analysis confirmed that four GRFs (*GRF1*, c56908.graph_c0; *GRF4*, c52862.graph_c0; *GRF6*, c39378.graph_c0 and *GRF9*, c54658.graph_c0) are *hpo-miR396b* targets that are regulated by specific cleavage at the binding site between the 10th and 11th nucleotides from the 5′ terminus of *hpo-miR396b*. Furthermore, quantitative real-time polymerase chain reaction (qRT-PCR) analysis showed that *hpo-miR396b* is down-regulated when confronted with drought stress (15% polyethylene glycol, PEG), and its expression fluctuates under other abiotic stresses, i.e., low temperature (4 ± 1 °C), high temperature (42 ± 1 °C), NaCl (100 mM), and abscisic acid (ABA; 0.38 mM). Conversely, the expression of *HpGRF6* showed the opposite trend to exposure to these abiotic stresses. Taken together, *hpo-miR396b* plays a regulatory role in the control of *HpGRF6*, which might influence the abiotic stress response of pitaya. This is the first documentation of this role in pitaya and improves the understanding of the molecular mechanisms underlying the tolerance to drought stress in this fruit.

## 1. Introduction

miRNAs are endogenous, single-strand, non-coding, small-molecular-weight RNAs of about 20 nucleotides (nt) in length, whose precursors are characterized by a stem loop structure [1,2]. They are known to regulate gene expression at the post-transcriptional level through cleavage and/or the translational repression of mRNAs [3]. miRNAs may also cause epigenetic modifications, including DNA and histone methylation, to control their targets [4,5,6]. Available evidence also justifies the idea that miRNAs are highly conserved, temporally and tissue-specifically expressed, and lead to the regulation of many biological processes, including those involved in growth, development, and metabolism, as well as abiotic and biotic stress responses in plants [7,8]. The post-transcriptional regulation of plant miRNAs to their targets is the main regulator of their function [9,10,11]. Thus, elucidation of the expression patterns of miRNAs and their corresponding targets at specific growth stages or in certain growth environments is of great significance for understanding their function [12].

Previous studies have demonstrated that in *Medicago truncatula*, *miR398* and *miR408* are induced by drought stress; conversely, the expression levels of the targets *Copper superoxide dismutase 1* (*CSD1*) and *Cytochrome coxidase subunit Vb* (*COX5b*) decrease significantly under such conditions [13]. The expression of the *miR169* is inhibited by low nitrogen levels in *Arabidopsis thaliana*, and its target *Nuclear transcription factor Yalpha* (*NFYA*), which is part of a network that regulates nitrogen metabolism in combination with *Nitrate transporters1.1* (*AtNRT1.1*) and *AtNRT2.1* in *A. thaliana,* was up-regulated [14]. More recently, it was found that *miR396* might regulate cell proliferation and differentiation by targeting growth-regulating factors (GRFs), thereby affecting growth and development and increasing a plant’s stress tolerance [15].

*GRF1*, which was first discovered in *Oryza sativa* (*Os-GRF1*), is a gibberellin-regulated transcription factor that was expressed in *Arabidopsis*, and the stem elongation of the transformed plants was greatly inhibited [16]. Subsequently, nine members of the *GRF* family were discovered in *A. thaliana*, whose domains are similar to those of *O. sativa* GRFs [17]. In general, the Gln–Leu–Gln (QLQ) and Trp–Arg–Cys (WRC) motifs are conserved in the N-terminals of *GRF* genes. The QLQ domain can affect the expression of target genes bearing the QLQ domain binding sequence, and the WRC region contains a nuclear localization signal and a DNA binding region [18]. In 2004, the *miR396* family was discovered by high-throughput sequencing (Illumina) of *A. thaliana*, and six *AtGRF* members were confirmed to be targets of *ath-miR396* by 5′-Rapid amplification of cDNA ends (5′-RACE) [19,20]. Various studies have proven that the regulation of GRFs by *miR396* plays an important role in plant growth and development and involves a variety of stress responses [21].

Pitaya (*Hylocereus polyrhizus*), belonging to the family Cactaceae, is an economical and nutritional fruit cultivated in tropical and subtropical regions, that is characterized by its high tolerance to drought stress [22,23]. Unfortunately, there are some limitations in its yield potential, namely drought and other adverse environments constrain its growth and development. Recently, the molecular response of pitaya to drought and salt stress at the transcriptomic level was documented [24,25]; however, the roles of miRNAs in this response have not yet been deciphered.

Previously, small RNA-seq, RNA-seq, and degradome-seq were applied in pitaya under exposure to drought stress, from which *hpo-miR396b*, whose target is *HpGRF6*, was found to be differentially expressed under drought stress. To unravel the roles of *miR396b* in response to abiotic stresses, in the current study, the hairpin sequence of *hpo-miR396b* and its target *HpGRF6* were isolated from pitaya. Supported by degradome sequencing, *HpGRF6* was found to be sliced by *hpo-miR396b* at specific sites. Subsequently, the expression levels of *hpo-miR396b* and its target *HpGRF6* under various stresses were determined. The results generated herein may further elucidate the roles of *miR396b* in plants’ adaptation to abiotic stresses.

## 2. Results

### 2.1. Cloning and Sequence Analysis of miR396b in Pitaya

The expected band at about 240 bp was observed by 2% agarose gel electrophoresis (Figure 1A), which is based on the stem-loop structure of *hpo-miR396b*. The sequencing result showed that the complete hairpin sequence was successfully cloned (130 bp). The alignment result showed that a similarity between the cloned hairpin and corresponding unigene of 100% (Figure 1B).

The RNA fold analysis of the hairpin sequence revealed that it forms a typical stem-loop structure with *hpo-miR396b* and *hpo-miR396b** located in the two opposite arms (shown by the red and purple line in Figure 2A). The *hpo-miR396b* sequence (5′-UUCCACAGCUUUCUUGAACUU-3′) is located in the 5p arm of pitaya precursor *miR396b* (*MIR396b*). Additionally, the *hpo-miR396b** read was uncovered from the RNA-seq data. Comparing the stem-loop structure of *H. polyrhizus*, *A. thaliana*, and *Nicotiana tabacum MIR396b*, the sequence of the two arm structure was found to be relatively conserved (Figure 2); however, their loops varied in sequence and minimum free energy (*H. polyrhizus* loop dG = −4.20 kcal/mol, *A. thaliana* loop dG = −9.4 kcal/mol, *N. tabacum* loop dG = −0.7 kcal/mol).

### 2.2. Phylogenetic Analysis of Plants MIR396b

The precursors of the miRNA genes are much longer than the mature miRNA molecule, and their nucleotide sequences vary within as well as between species. Therefore, phylogenetic analysis of the precursor sequences of a MIRNA family may reveal the true evolutionary relationships between its MIRNA genes. Thirty-four species of plant *MIR396b*, including pitaya, were identified in this study through phylogenetic analysis. Based on the phylogenetic tree (Figure 3), the whole system was divided into three branches, and the monocotyledonous *Vriesea carinata*, *Aegilops tauschii*, and *Brachypodium distachyon* were clustered into a branch. The gymnosperm plant of *Picea abies* was separately clustered into a single branch, whereas the monocotyledonous *Zea mays*, *Sorghum bicolor*, *Asparagus officinalis*, and *Oryza sativa* and dicotyledons including pitaya *MIR396b* were in another branch. The pitaya and dicotyledonous plants were more closely clustered than the plant of monocotyledonous and gymnosperm, and the *MIR396b* sequences of members of the Brassicaceae, Poaceae, and Solanaceae families and other plants were clustered together. Plant *MIR396b* has a more recent evolutionary relationship in the family, but there were also a few species with intra-family plant aggregation, such as legume plants, like *Glycine max* and *Medicago truncatula*. The clustering of pitaya and the dicotyledonous species was more scattered, which further demonstrated that, in addition to species differences, there are other factors that influence the formation of *MIR396b*, also indicating that the origin of the pitaya *MIR396b* is quite special.

### 2.3. Conservation Analysis of Plants miR396b

Alignment between *miR396b* sequence of twelve plant species including *hpo-miR396b* analyzed from the miRbase database (Release 22.1) (Figure 4A), and the mature sequence of *hpo-miR396b* was consistent with most of the other plants’ *miR396b*. The substitution analysis of the 5′ product of *miR396b* showed that its bases were highly conserved (Figure 4B), indicating that it is of great importance and is highly conserved during evolution.

### 2.4. Prediction of Hpo-miR396b Target Genes

To date, all *miR396*-targeted genes identified from *A. thaliana* have been found to belong to the *GRF* gene family. To identify *hpo-miR396b* targets in pitaya, pitaya mRNA sequences were searched, and complementary sequences to the mature *hpo-miR396b* sequences were found based on the near-perfect complementarity principle. The number of predicted targets of *hpo-miR396b* is 19, of which 17 have been annotated (Appendix A
Table A1). The preliminary annotations suggest that *hpo-miR396b* is involved in a variety of biological processes such as transcriptional regulation, substance metabolism, and stress response. Supported by degradome sequencing (SRA: SRR8767413), four GRFs (*GRF1*, c56908.graph_c0; *GRF4*, c52862.graph_c0; *GRF6*, c39378.graph_c0 and *GRF9*, c54658.graph_c0) were shown to be *hpo-miR396b* targets that are regulated by specific cleavage at the binding site between the 10th and the 11th nucleotides from the 5′ terminus of *hpo-miR396b* (Figure 5).

### 2.5. Cloning and Structure Analysis of HpGRF6

The open reading frame (ORF) of target gene *HpGRF6* (c39378.graph_c0) was amplified. The expected band at about 1200 bp was detected by 2% agarose gel electrophoresis (Figure 6A). The sequencing result showed that the cloned sequence is 1154 bp and contains 1119 bp ORF encoding a polypeptide chain of 372 amino acids (GeneBank: MK625692). Smart software predicted the domain and demonstrated its similarity to *GRF* genes in other plants, whose conserved domains are QLQ and WRC. The sequence analysis indicated that the interaction site in *miR396b-HpGRF6* is located in the WRC domain of the coding region ‘CCGUUCAAGAAAGCCUGUGGAA’, coding the conserved “RSRKPVE” amino acid sequence (Figure 6B). BoxShade was used to compare the transcription factor GRFs of *H. polyrhizus*, *Chenopodium quinoa*, and *A. thaliana*, and the similarities with *C. quinoa* and *A. thaliana* were 98% and 91% (Figure 6C), respectively, indicating high conservation among the three GRFs. In addition, *miR396* was shown to be conserved.

### 2.6. Subcellular Localization of HpGRF6 Protein

The ORF of *HpGRF6* without a stop codon was inserted into the pBWA(V)HS-GLosgfp vector to express the pBWA(V)HS-HpGRF6-GLosgfp (control green fluorescence protein (GFP)) recombinant protein, and the constructed plasmid vector was transformed into tobacco. A fluorescence signal was found in the nuclear region of epidermal cells (Figure 7), justifying that the WRC region of the *HpGRF6* gene has the same nuclear localization signal as *GRF* in other species.

### 2.7. Expression of hpo-miR396b and HpGRF6 during Exposure to Abiotic Stresses

In order to check the negative correlation between *hpo-miR396b* and *GRF*, we analyzed the expression of *hpo-miR396b* and its target gene *HpGRF6* following exposure to treatments with polyethylene glycol (PEG), high/low temperature, NaCl, and abscisic acid (ABA). The expression of *hpo-miR396b* was obviously down-regulated under drought stress, obtaining a minimum level (−37%) after 20 h (Figure 8A), while the expression of *HpGRF6* showed the opposite trend, and reached a maximum (+5.4-fold) at 20 h after treatment with 15% PEG. *hpo-miR396b* and *HpGRF6* expression fluctuated under both high and low temperatures, reaching the highest point at 8 h (*p* < 0.01) after exposure to high-temperature stress (Figure 8B); however, the expression of *hpo-miR396b* rapidly and transiently increased after exposure to low temperature for 2 h (Figure 8C), and then decreased at 8 h, reaching its lowest point (*p* < 0.05). We also analyzed the *HpGRF6* gene under the same conditions; its expression showed the exact opposite trends to *hpo-miR396b* (Figure 8C). After 20 h of treatment with NaCl (300 mM), *hpo-miR396b* was responsive to salt stresses and reached its maximum expression at 2 h, before declining until 20 h (Figure 8D). *Hpo-miR396b* was also involved in the response to ABA stresses, but the differences did not reach a significant level (Figure 8E). Conversely, under both NaCl and ABA stresses, *HpGRF6* showed a trend that was opposite to that of *hpo-miR396b*, suggesting that *HpGRF6* is involved in slat and phytohormone ABA stresses. Moreover, the expression of *hpo-miR396b* and *HpGRF6* showed opposite trends under exposure to these abiotic stresses, suggesting that *HpGRF6* is the *hpo-miR396b*-targeted gene, demonstrating the crucial role it may play in the plant stress response as well as in plant defense.

To understand the spatio-temporal expression of *miR396b*, quantitative real-time polymerase chain reaction (qRT-PCR) was carried out and the results showed that *hpo-miR396b* is expressed differentially in pitaya tissues (Figure 8F). Using the stem as a control, expression was barely detected in the root, while it was about 26 times higher in the fruit, reaching an extremely significant level (*p* < 0.01).

## 3. Discussion

### 3.1. Evolution and Conservation Analysis of Plants miR396b

miRNA genes were widely distributed in plants such as trees, herbage, or even bryophytes, etc. Three hypotheses had been proposed to explain their origins, of which the first was that the new miRNA gene originated from a duplication of genetic elements such as miRNA and protein-coding genes, and the second was that terminal inverted repeats of transposable elements became miRNA genes [26,27,28]. During the past decade, some research suggested that most miRNAs originated from random hairpin (stem-loop) structures, which was the third hypotheses [29,30], because there are hundreds of thousands hairpin structures in the genomes of higher organisms and some of which may become miRNA genes [31]. Additionally, lots of miRNA genes identified by these efforts were not related to the phylogeny but rather to the tissue type and developmental stages of plants. In liverwort, 428 miRNAs had been obtained from *Pellia endiviifolia* [32]; however, only 129 were available from *Marchantia polymorpha* [33]. Interestingly, our results show that there was a typical stem-loop structure in *MIR396b* of pitaya, indicating that it might become *miR396b* subsequently. No *MIR396b* in liverworts or algae was found in the miRBase (Release 22.1), which might be ascribed to the tissue- or species-specific expression. To examine the conservation of *miR396b* genes, alignment of *miR396b* sequences from twelve plant species was carried out (Figure 4A). The sequence of *hpo-miR396b* was identical to *A. thaliana*, *N. tabacum*, and *G. max*, etc., and showed one base difference with *Picea abies*. The nucleotide substitution analysis for the 5′ product of *miR396b* demonstrated that its sequence was highly conserved (Figure 4B).

### 3.2. Target Genes of miR396b

Unlike functional genes, miRNA itself does not translate into proteins; however, it may regulate target genes. Therefore, to better unravel the function of miRNA, we investigated the target genes of its action. At present, degradome sequencing has been used to identify miRNA cleavage sites [34], because miRNAs can cause the endonucleolytic cleavage of mRNA by extensive and often perfect complementarity to mRNAs [35]. Degradome sequencing has revealed many known and novel plant miRNA targets [36,37]. Recently, it has also been applied to identify *A. thaliana* [38], *Z. mays* [39], and *Gossypium hirsutum* [40], among other miRNA-derived cleavages. Shamimuzzaman (2012) obtained 183 target genes from 53 conserved miRNAs by analyzing the soybean seed degradome sequencing library [41]. Potential target genes for some conserved and novel miRNAs were identified by degradome sequencing in peanuts [42].

To date, there has been no research into the cactus family’s miRNA or the understanding of its target genes. Recently, investigations have shown that *miR396*-mediated GRFs may play a role in coordinating plant growth through defense signaling and stress responses [43,44]. In addition, the basic helix–loop–helix *(bHLH*) transcription factor 74 gene (*bHLH74*) was identified as the target of *miR396* that acts as a regulator for root growth in *Arabidopsis* seedlings [45,46], but *bHLH74* homologs with a *miR396* target site were only detected in the sister families Brassicaceae and Cleomaceae [47]. In the present study, a detailed predictive analysis of the *hpo-miR396b* target genes was carried out. Based on the findings herein, the main target genes of *hpo-miR396b* are also members of the *HpGRF* family. Supported by degradome sequencing, four of the GRFs were shown to be *hpo-miR396b* targets, and they are all regulated by specific cleavage at the binding site between the ‘CCGUUCAAGAA’ and the ‘AGCCUGUGGAA’ nucleotides from the 5′ terminus of *HpGRFs*, which is consistent with the cleavage site reported in tea [48]. It was further shown that *hpo-miR396b* can cleave the targets at the transcriptional level to regulate their expression.

### 3.3. Tissue-Specific and Abiotic Stress Response of miR396 in Plant

As a conserved and well-studied miRNA family in terrestrial plants, *miR396* is known to be expressed in different plant tissues. Rodriguez et al. (2010) and Bao et al. (2014) detected the expression of *miR396* in the leaves and roots of *Arabidopsis* [15,46], and Baucher et al. (2012) found that *miR396* is expressed in the floral organs of poplar [49]. Available evidence also suggests that *mtr-miR396a* and *mtr-miR396b* genes are highly expressed in the tips of roots and display distinct expression profiles during lateral root and nodule development in *Medicago truncatula* [50]. In the present study, the expression of *hpo-miR396b* in pitaya fruit was shown to be significantly higher than that in other organs, indicating that the expression pattern of *hpo-miR396* is tissue-specific.

Unlike animals, which are capable of escaping stressful conditions by moving away, plants must cope with stress directly at the place where the seed has germinated. As a consequence, the growth of plants is often influenced by stress. A great deal of research has shown that the activation of stress-related genes increases a plant’s tolerance to stress [51,52]. In recent studies, it was demonstrated that *miR396* is involved in responses to a variety of biotic and abiotic stresses. Liu et al. found that transgenic *miR396*-overexpressing plants are more tolerant to drought than wild-type plants in *Arabidopsis* [53]. In particular, the *AtGRF3* transgene yielded an increase in leaf size under mild drought stress and showed enhanced resistance to certain plant pathogens [54]. Further analysis revealed that *Sp-miR396a-5p* transcription levels are up-regulated under salt and drought stresses, and additionally, the expression of *Sp-miR396a-5p* is down-regulated under pathogen-related biotic stress [55], and the relevant link between salt and alkali stress of *osa-miR396c* in rice was established by Gao et al. [56]. Recent clues also revealed that homeostasis in the *miR396-GRF* regulatory network was essential for productive *Heterodera glycines* infections in soybean [57]. Considering all evidence, it is clear that *miR396*-targeted GRFs play important roles in the response to biotic and abiotic stresses, including drought, salt, alkali, and UV-B radiation [58], among others. Interestingly, our results showed that the expression of *hpo-miR396b* and *HpGRF6* have opposite trends, which reach different levels during responses to abiotic stresses in pitaya. This might be helpful for further elucidating the roles of *miR396b* in plants’ adaptation to stress tolerance. For the first time, miRNA and mRNA expression were analyzed during the osmotic stress response in pitaya, giving new insight into the regulation of pitaya’s adaptation to abiotic stress tolerance.

## 4. Materials and Methods

### 4.1. Plant Material and Stress Treatment

Given that the stem is the primary organ of pitaya seedlings, it was used to examine differential expression of *miR396b* and its target in response to abiotic stresses. The stem of pitaya (*H. polyrhizus*) was obtained from the research greenhouse, Ministry of Education, Institute of Agro-Bioengineering, Guizhou University. To verify the expression of the *hpo-miR396b’* targeted *HpGRF6* gene under various abiotic stresses and phytohormone treatments, selected aseptic seedlings of pitaya were cultured in 1/2 MS liquid medium at a day/night temperature of 25 ± 1/20 ± 1 °C inside a greenhouse, with 50–60% relative humidity. After 1 week, the aseptic seedlings were transferred into 1/2 MS liquid medium containing 15% PEG-8000/100 mM NaCl solution and exposed to artificially induced drought/salt stress treatments for 0, 2, 8, and 20 h. Some of the seedlings were transferred into 1/2 MS liquid medium containing ABA (0.38 mM), and exposed to artificially induced hormone stress treatments for 0, 8, and 20 h. Untreated material was used as a control. For cold/heat stress, seedlings were transferred to the artificial climate incubator for 4 ± 1 °C/42 ± 1 °C treatments for 0, 2, 8, and 20 h, with normal-temperature (25 ± 1 °C) material as a control.

To investigate the tissue-specific expression of *miR396b*, samples representing the tissues of the stem, root and fruit of pitaya (*H. polyrhizus*), were collected according to the designated quantitative expression level analysis, which were provided by the Institute of Fruit Trees, Guizhou Province Academy of Agricultural Sciences.

### 4.2. Prediction of Hpo-miR396b Target Genes in Pitaya

The sequences of *hpo-miR396b* and mRNA were obtained from our previous small RNA library, and RNA-Seq were constructed from the stems of pitaya conducted with drought stress high-throughput sequencing and input into psRNATarget (http://plantgrn.noble.org/psRNATarget/, 2017 Release) and Targetfinder (http://targetfinder.org/, 2010 Release) to search for targets in pitaya [59,60]. Its library maximum expectation of ≤4.0 was selected for target prediction. We took the intersection of the software psRNATarget and Targetfinder predictions as the final result, supported by the degradome sequencing of pitaya in drought stress to identify the predicted targets.

### 4.3. Gene Cloning and Sequence Analysis

To form the gene clone of hairpin sequence of pitaya *miR396b* by PCR, whose primers (Appendix A
Table A2) were designed based on the small RNA sequences, DNA was isolated using DNAsecure Plant Kit (Tiangen, Beijing, China) at a total volume of 20 μL containing 10 μL of Taq PCR Master Mix (Tiangen), 1 μL of each primer, 1 μL gDNA, and 7 μL of ddH_2_O. PCR conditions were as follows: 95 °C for 10 min followed by 35 cycles of 95 °C for 30 s and 60 °C for 1 min. Obtained PCR products were resolved in 2% agarose gels, according to standard procedures. The target fragment was purified by TIANgel Midi Purification Kit (Tiangen) and then it was cloned into pMD19-T (Takara, Dalian, China), yielding the plasmid pMD19-T-miR396b. Next, the ligated vector was transformed into DH5α competent cells, which were cultured overnight in LB medium containing Amp, and positive clones were selected for sequencing.

Based on the results of the prediction and degradome sequencing, *HpGRF6* (c39378.graph_c0) was selected to design the primer (Appendix A
Table A2) containing the full length of ORF. RNA was isolated using the miRcute miRNA Extraction and Isolation Kit (Tiangen). cDNA was synthesized by the RevertAid First Strand cDNA Synthesis Kit (Thermo Scientific, Waltham, MA, USA), where 2 μg of total RNA was reverse-transcribed in a 20 μL reaction containing oligo-dT primer 1 μL, Random Hexamer Primer 1 μL, 5× Reaction Buffer 4 μL, 10 μM dNTPs (mix) 2 μL, RiboLock RNase Inhibitor 1 μL, and ReverAid M-MuLV RT 1 μL, at 42 °C for 60 min, and heat inactivation of the enzyme was performed at 70 °C for 5 min. The method followed was the same as that used for the cloning of the hairpin *hpo-miR396b* sequence.

### 4.4. Bioinformatics Analysis

Both mature and hairpin (precursor) sequences of plant *miR396b* were downloaded from the miRBase sequence database (http://www.mirbase.org, Release 22.1). Conserved amino acid sequences were determined by multiple sequence alignment (MSA) using Clustal X (Version 1.81) and BoxShade (http://sourceforge.net/projects/boxshade, version 1.8) tools. The RNAfold Web Server (http://unafold.rna.albany.edu, version 3.4) predicted the secondary structure of hairpin sequence *hpo-miR396b*. A phylogenetic tree of plant *MIR396b* was generated by MEGA (Tokyo Metropolitan University, Tokyo, Japan, version 6.0) using the neighbor-joining method. The conservation of plant *miR396b* was analyzed by Weblogo (http://weblogo.berkeley.edu/, version 2.8). The ORF of *HpGRF6* was analyzed by NCBI (https://www.ncbi.nlm.nih.gov/orffinder/, National Center for Biotechnology Information, U.S. National Library of Medicine), and SMART software (http://smart.embl-heidelberg.de/, version 7.0) predicted the conserved domain. Subcellular localization of *HpGRF6* protein was predicted using TargetP (http://www.cbs.dtu.dk/services/TargetP/, version 1.1)

### 4.5. HpGRF6 Protein Subcellular Localization

The subcellular localization expression of *HpGRF6* protein was observed by the tobacco leaf transient transformation technique. The ORF without a stop codon of *HpGRF6* (Appendix A
Table A2) was inserted into the pBWA(V)HS-GLosgfp vector, which was provided by BIORUN Technologies company (Wuhan, China) to express pBWA(V)HS-HpGRF6-GLosgfp (control GFP) recombinant protein. In addition, transformation analysis was performed according to the transient transformation scheme of tobacco leaf epidermal cells [61]. Transfected genes of interest were observed under a confocal laser scanning microscope C2-ER, (Nikon, Tokyo, Japan) to determine the expression of GFP in tobacco cells.

### 4.6. Quantitative Real-Time Polymerase Chain Reaction (qRT-PCR) Analysis

For the qRT-PCR analysis of *HpGRF6* and *hpo-miR396b*, total RNA was isolated using the miRcute miRNA Extraction and Isolation Kit (Tiangen), followed by the RNA quality and purity measurement by the NanoDrop 2000 (Thermo Scientific, Waltham, MA, USA). cDNA was synthesized by the RevertAid First Strand cDNA Synthesis Kit (Thermo Scientific). qRT-PCR was performed on the Real-Time PCR system (Applied Biosystems, Foster City, CA, USA) using the SYBR Green Master Mix, with no less than three independent biological replicates, each comprising three technical replicates. Each technical replicate was prepared in a total volume of 20 μL containing 10 μL of SYBR Green PCR Master Mix, 1 μL of each primer, 2 μL cDNA, and 6 μL of ddH_2_O. Stem-loop qRT-PCR for mature *miR396b* was conducted as described previously [62,63], with *U6*/*RP40S* as the pitaya *miR396b*/*HpGRF6* endogenous reference, respectively. The qRT-PCR conditions were as follows: 95 °C for 5 min followed by 35 cycles of 95 °C for 30 s and 60 °C for 30 s. The primers (Appendix A
Table A2) were designed by Primer Premier 5.0.

### 4.7. Statistical Analysis

The preceding Section 4.6 qRT-PCR analysis, all experiments were repeated three times, and all data were analyzed by Duncan’s multiple range test following two-way ANOVA analysis. Statistical analyses were carried out using Microsoft Excel (Version 2007, Microsoft Corporation, Washington, DC, USA) and SPSS (Version 21.0, International Business Machines Corporation, New York, NY, USA) software, and statistical significance was taken as *p* < 0.05. The relative expression level was calculated with the formula 2^−ΔΔ*C*t^ = normalized expression ratio [64].

## Figures and Tables

**Figure 1 ijms-20-02501-f001:**
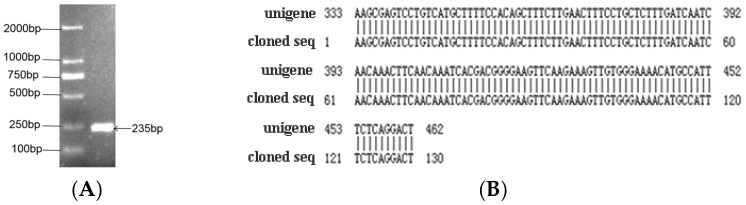
Cloning and sequence analysis of pitaya hairpin sequence *MIR396b*: (**A**) electrophoresis of cloned *hpo-miR396b* precursor; (**B**) alignment of the clone with the transcriptome.

**Figure 2 ijms-20-02501-f002:**
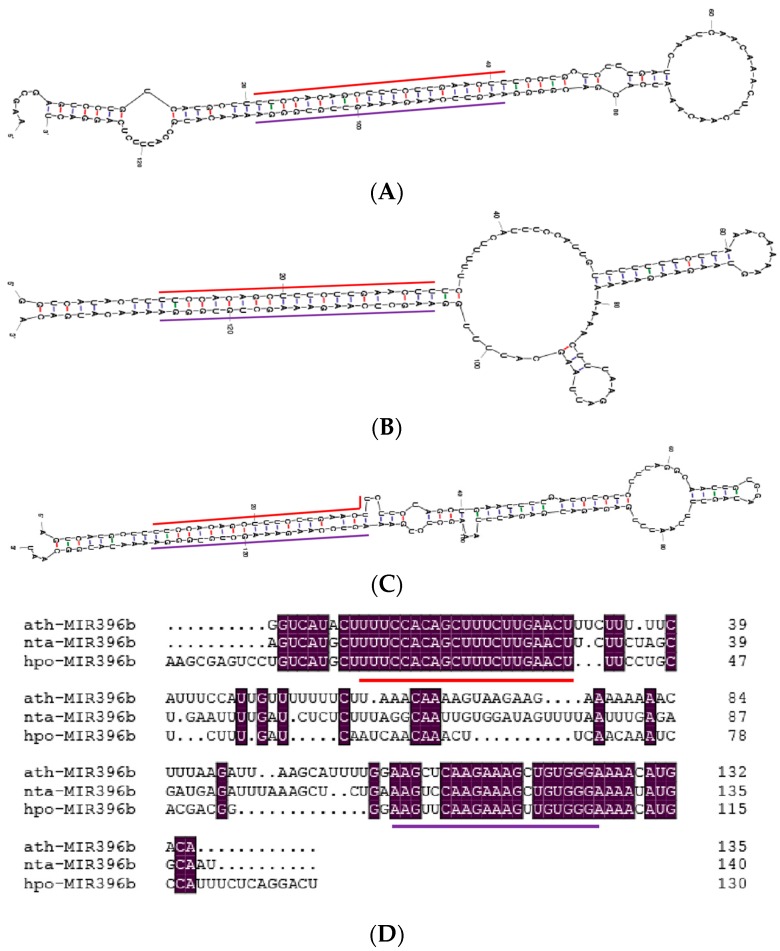
The comparison of the stem-loop structure of *miR396b* among three plant species: (**A**) *H. polyrhizus* (minimum free energy dG = −55.80 kcal/mol); (**B**) *A. thaliana* (minimum free energy dG = −43.59 kcal/mol); (**C**) *N. tabacum* (minimum free energy dG = −55.90 kcal/mol); (**D**) Alignment sequence analysis of *MIR396b* from the three species. *miR396b* and *miR396b** are shown by the red and purple line in Figure 2. Nucleotides that are identical are highlighted in purple background.

**Figure 3 ijms-20-02501-f003:**
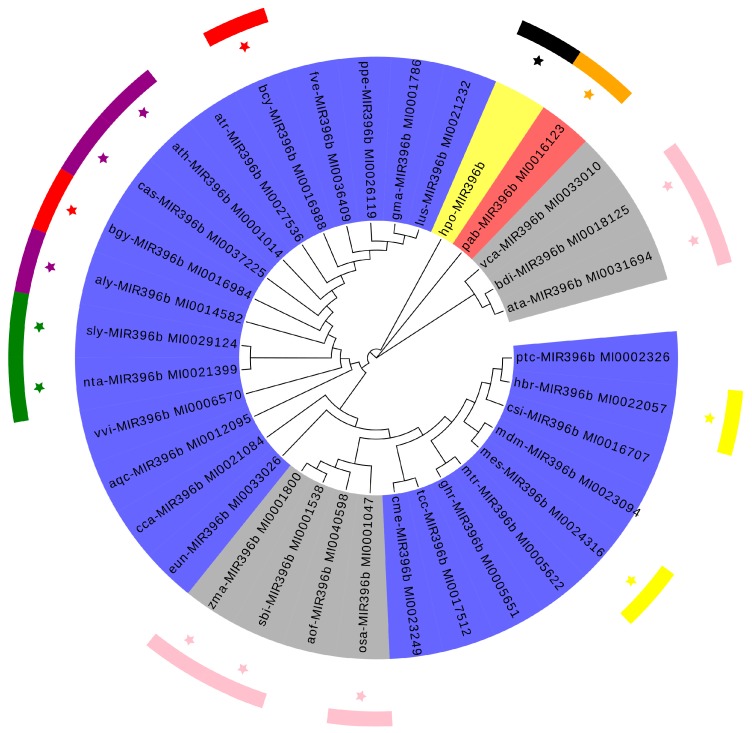
Phylogenetic tree of *MIR396b* from thirty-four plant species. MEGA (Tokyo Metropolitan University, Tokyo, Japan, Version 6.0) was used to build the neighbor-joining (NJ) tree with 1000 bootstrap replicates. Different colors indicate different classes of plants. Blue, gray, and red represent the plant dicotyledons, monocotyledonous, and gymnosperm, respectively. *H. polyrhizus* is represented by yellow. In addition, different star-line colors indicate different families of plants.

**Figure 4 ijms-20-02501-f004:**
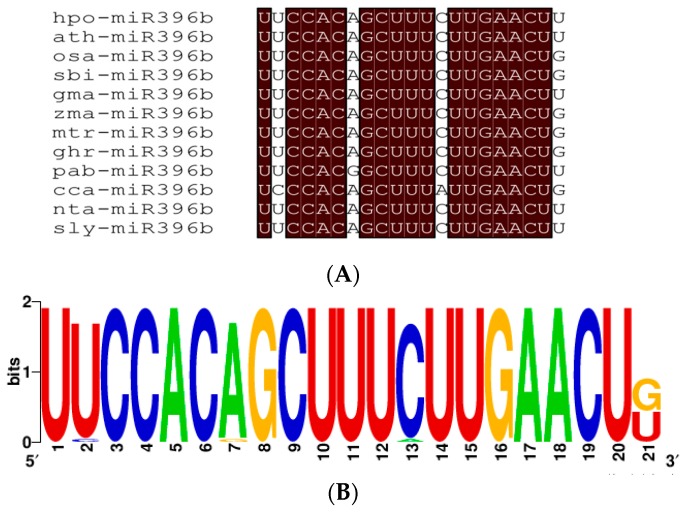
Conserved nucleotide sequence analysis of plant *miR396b*: (**A**) alignment of the logo sequence of miR396b from twelve plant species whose names are *H. polyrhizus* (*hpo-miR396b*), *A. thaliana* (*ath-miR396b*), *O. sativa* (*osa-miR396b*), *Sorghum bicolor* (*sbi-miR396b*), *Glycine max* (*gma-miR396b*), *Zea mays* (*zma-miR396b*), *Medicago truncatula* (*mtr-miR396b*), *Gossypium hirsutum* (*ghr-miR396b*), *Picea abies* (*pab-miR396b*), *Cynara cardunculus* (*cca-miR396b*), *Nicotiana tabacum* (*nta-miR396b*), and *Solanum lycopersicum* (*sly-miR396b*), respectively. Nucleotides that are identical are highlighted in dark red background; (**B**) the logo sequence of *miR396b*, and the height of the letter at each position represents the degree of conservation.

**Figure 5 ijms-20-02501-f005:**
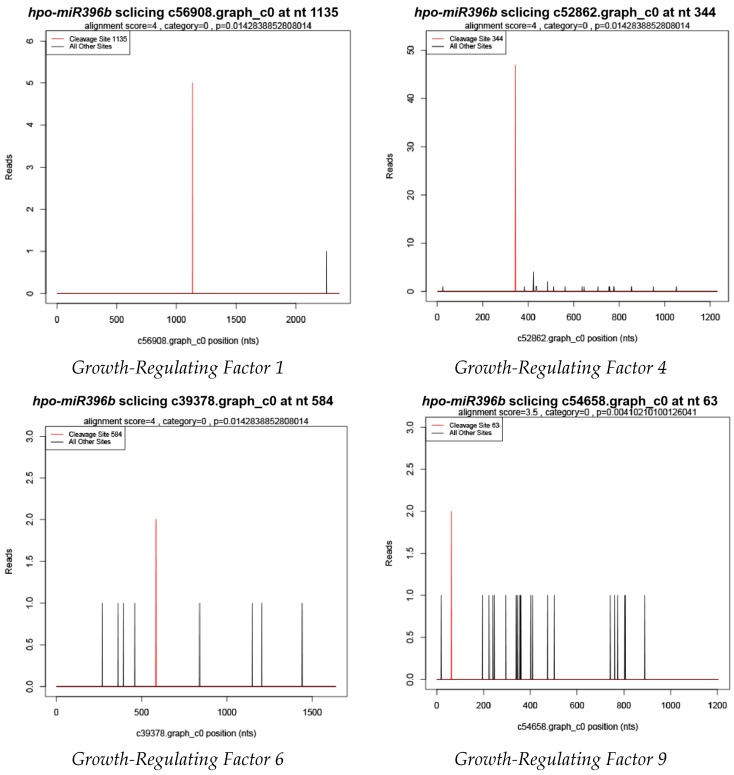
Target plot (t-plot) of target genes of the growth regulating factor (*GRF*) of *hpo-miR396b* gained by degradome sequencing. The red lines of t-plots indicate that the specific cleavage site. The *X* axis indicates the unigene. The *y* axis indicates the raw reads.

**Figure 6 ijms-20-02501-f006:**
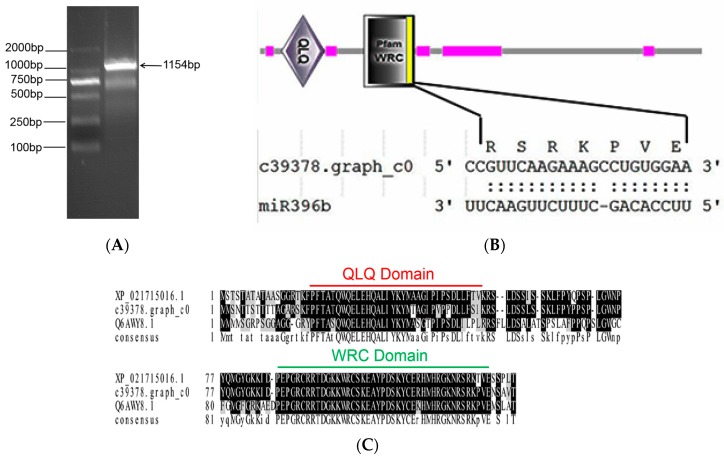
Cloning and sequence analysis of *HpGRF6*: (**A**) electrophoresis of cloned *HpGRF6*; (**B**) prediction of the conservative domain of *HpGRF6*, whose conserved domains are QLQ and WRC, and the double black points indicate matched base pairs; (**C**) amino acid sequence alignment of *HpGRF6* with the conserved domains of other plant species. XP_021715016.1, Q6AWY8.1 and c39378.graph_c0 represent the plants *C. quinoa*, *A. thaliana*, and *H. polyrhizus*, respectively. Residues that are identical are highlighted in black and different residues are highlighted in grey background.

**Figure 7 ijms-20-02501-f007:**
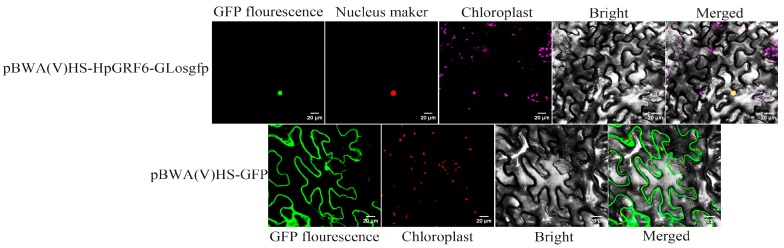
Subcellular localization of *HpGRF6* in tobacco leaves. The construct of *HpGRF6* fused with the N-terminal of green fluorescence protein (GFP; pBWA(V)HS-HpGRF6-GLosgfp) was co-infiltrated with pBWA(V)HS-GFP into tobacco leaves. The co-infiltrated leaves were photographed after 3 days. Note: GFP fluorescence (green), nucleus fluorescence (red), merged images (yellow), and bright-field chloroplast (rose red) images are shown. Scale bar = 20 μm.

**Figure 8 ijms-20-02501-f008:**
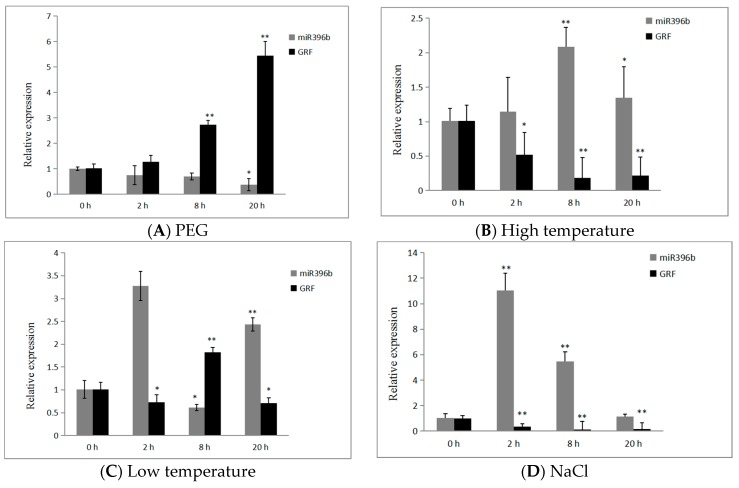
Expression patterns of *hpo-miR396b* and the target gene *HpGRF6* under exposure to different treatments: (**A**–**E**) expression of *hpo-miR396b* and *HpGRF6* induced by polyethylene glycol (15% PEG-8000), 42 ± 1 °C, 4 ± 1 °C, NaCl (100 mM) and abscisic acid (ABA; 0.38 mM); (**F**) expression of *hpo-miR396b* in different tissues. All data were analyzed by two-way ANOVA, and followed by Duncan’s multiple range test. “*” indicates a significant difference at the level of *p* < 0.05. “**” indicates a significant difference at the level of *p* < 0.01.

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
