# Peer review of "Conserved miR396b-GRF Regulation Is Involved in Abiotic Stress Responses in Pitaya (Hylocereus polyrhizus)"

_ijms, 2019, doi:10.3390/ijms20102501_

Reviewer 1 Report

The paper entitled “Conserved miR396b Targeting GRF Genes Is Involved in the Abiotic Stress Response of Pitaya (Hylocereus polyrhizus)” describes the miR396 and its target in cactus. Moreover there is an information about changes in expression level of miR396/GRF upon abiotic stresses. The paper could be published but in my opinion only when the authors put more stress on the evolutionary position of miR396 and its target. The similarity and differences to other plants should be discussed. So it will be interesting to include in the analysis such plants like liverworts, mosses or even algae. Author wrote that their paper describes for the first time miR396/GFR in pitaya. So please write what do you have novel compared to the article written by Quandong Nong, Mingyong Zhang, Jiantong Chen et al., “RNA-Seq De Novo Assembly of Red Pitaya (Hylocereus polyrhizus) Roots and Differential Transcriptome Analysis in Response to Salt Stress” February 2019Tropical Plant Biology, DOI: 10.1007/s12042-019-09217-3. In the text you put information that … Previously, small RNA-seq, RNA-seq, and degradome-seq were determined in pitaya under exposure to drought stress, from which hpo-miR396b, whose target is HpGRF6, was found to be differentially expressed under drought stress. To… where it was published, what novel data are presented in the submitted paper.

Below I put notes which should be included in the revised manuscript:

1.      I am curious how the drought stress was controlled and analysed in cactus since this species is resistant to water deficiency. So I am not sure if 20 h exposure for PEG can mimic drought stress. Did you analyse dehydrin gene expression level to show that the plants really were stressed.

2.      How the degradome libraries were prepared? Did you submit the degradome-seq to the database?

3.      Page 2 - Be more precisely describing the role of the GRF1, whether it inhibits or stimulates stem growth;

4.      Page 2 - There are more recent papers about miR396-GRF interactions which should be included and discussed in the paper like:

Noon JB, Hewezi T, Baum TJ. Homeostasis in the soybean miRNA396-GRF network is

essential for productive soybean cyst nematode infections. J Exp Bot. 2019 Mar

11;70(5):1653-1668. doi: 10.1093/jxb/erz022

Beltramino M, Ercoli MF, Debernardi JM, Goldy C, Rojas AML, Nota F, Alvarez

ME, Vercruyssen L, Inzé D, Palatnik JF, Rodriguez RE. Robust increase of leaf

size by Arabidopsis thaliana GRF3-like transcription factors under different

growth conditions. Sci Rep. 2018 Sep 7;8(1):13447. doi:

10.1038/s41598-018-29859-9

5.      The expected band at about 240 bp… expected based on what pre-microRNA, pri-microRNA?

6.      Figure 2 - Please mark the position of microRNA and microRNA* also on pre-microRNA structures for Arabidopsis and tobacco. Moreover did you find in the RNA-seq data microRNA396* read?

7.      Figure 3 - Firstly prepare alignment between miR396 sequence from three species analyzed in Figure 2 and secondly describing the logo sequence in Figure 3 put information how many plant species (which) were analysed to create the logo sequence;

8.      Figure 4 - What kind of sequences were used for the tree creation: MIR gene, pri-microRNA, pre-microRNA, mature microRNA?

9.      Figure 5 - Once again be more precisely, x-axis this is nucleotide position within I guess cDNA? Or maybe DNA, y-axis this is what? normalized reads, raw reads, you should put dotted line showing on each graph average distribution of degradome fragments per nucleotide;

10.  Figure 7 - What kind of nucleus marker was used, why chloroplast are not red under uv light?

11.  Figure 8 – Why in panel A there is no GRF expression bars?

12.  Please edit your manuscript more carefully: analysis.Based, SorgHpm bicolor;

13.  Please explain why did you use term crop for pitaya;

Author Response

Point 1: I am curious how the drought stress was controlled and analysed in cactus since this species is resistant to water deficiency. So I am not sure if 20 h exposure for PEG can mimic drought stress. Did you analyse dehydrin gene expression level to show that the plants really were stressed.

Response 1: Thanks for your conment. In 2013, Fan et al., had demonstrated that dehydration responsive domain partial (E-value 4.3E-44) expressed differentially in pitaya under PEG treatment which proved that PEG can mimic drought stress, though no differential expression of dehydrin was observed. The article was quoted in the introduction in line 72 of the revised version ([24] Fan QJ, et al. Gene 2014, 533, 322-331).

Point 2: How the degradome libraries were prepared? Did you submit the degradome-seq to the database?

Response 2: The samples used for construction of degradome libraries were prepared by PEG-treated at different times and non-treated pitaya stems in equal proportions, and was performed by Biomarker Technologies company (Beijing, China). The raw data has been submitted to SRA under accession number SRR8767413.

Point 3Page 2. Be more precisely describing the role of the GRF1, whether it inhibits or stimulates stem growth.

Response 3: that was expressed in Arabidopsis, and found the stem elongation of the transformed plants was greatly inhibited” were added in lines 57-58 in the revised version.

Point 4: Page 2. There are more recent papers about miR396-GRF interactions which should be included and discussed in the paper like:

Noon JB, Hewezi T, Baum TJ. Homeostasis in the soybean miRNA396-GRF network is essential for productive soybean cyst nematode infections. J Exp Bot. 2019 Mar

11; 70(5):1653-1668. doi: 10.1093/jxb/erz022

Beltramino M, Ercoli MF, Debernardi JM, Goldy C, Rojas AML, Nota F, Alvarez

ME, Vercruyssen L, Inzé D, Palatnik JF, Rodriguez RE. Robust increase of leaf

size by Arabidopsis thaliana GRF3-like transcription factors under different

growth conditions. Sci Rep. 2018 Sep 7;8(1):13447. doi:10.1038/s41598-018-29859-9

Response 4: Two sentences, e.g. “In particular, AtGRF3 transgene gave an increase in leaf size under mild drought stress and showed the enhanced resistance to certain plant pathogens” and “Also, recent clues revealed that homeostasis in the miR396-GRF regulatory network was essential for productive Heterodera glycines infections in soybean” were added in lines 313-314 and 318-319 of the new version, respectively.

Point 5 The expected band at about 240 bp… expected based on what pre-microRNA, pri-microRNA?

Response 5: “which is based on the stem-loop structure of hpo-miR396b” were added in line 84 of the revised version.

Point 6: Figure 2. Please mark the position of microRNA and microRNA* also on pre-microRNA structures for Arabidopsis and tobacco. Moreover did you find in the RNA-seq data microRNA396* read?

Response 6: The position of miR396b and miR396b* were marked on pre-microRNA structures for Arabidopsis and tobacco in Figure 2 of the revised version. “Additionally, the hpo-miRNA396b* read was uncovered from the RNA-seq data.” were added in line 95-96 of the revised version.

Point 7: Figure 3. Firstly prepare alignment between miR396 sequence from three species analyzed in Figure 2 and secondly describing the logo sequence in Figure 3 put information how many plant species (which) were analysed to create the logo sequence.

Response 7: The alignment of MiR396b sequence from Hylocereus polyrhizus (hpo-MIR396b), Arabidopsis thaliana (ath-MIR396b), and Nicotiana tabacum (nta-MIR396b) the three species was added as Figure 2D in the revision. Additionally, alignment of miR396b from twelve plant species in miRbase was added as Figure 1A. Names of sequences were added in the figure legend.

Point 8: Figure 4. What kind of sequences were used for the tree creation: MIR gene, pri-microRNA, pre-microRNA, mature microRNA?

Response 8: MIR396b sequences from different plant species were used for the tree creation.

Point 9: Figure 5. Once again be more precisely, x-axis this is nucleotide position within I guess cDNA? Or maybe DNA, y-axis this is what? normalized reads, raw reads, you should put dotted line showing on each graph average distribution of degradome fragments per nucleotide.

Response 9: The X axis indicates coordinates of Unigene, and the Y axis indicates the raw reads. We made the corresponding revision of the sentence in Figure 5. in the new version. Additionally, we are very sorry to tell you that we can't understand the meaning of the sentence “you should put dotted line showing on each graph average distribution of degradome fragments per nucleotide.” So we have no idea of the answer to the problem. Can you explain it? We’d like to make the further corresponding revision if it is necessary.

Point 10Figure 7. What kind of nucleus marker was used, why chloroplast are not red under uv light?

Response 10: Nucleus marker: mkate,

excitation light at 561nm,

emission light at 580 nm.

The chloroplast is red fluorescence at 640 nm under uv light, and the nuclear marker is also red at 561 nm. The two channels do not interfere with each other when photographing, and the original file is red, and but when the exported image of the two reds will be confused in merged. So the chloroplast fluorescence is changed to rose red in Figure 7. panel Choroplast, and in merged the choroplast also is rose red.

Point 11: Figure 8. Why in panel A there is no GRF expression bars?

Response 11: Thanks for your advice, we revised the corresponding part of the main text and the reason that no GRF expression bars were given is explained below. Negative correlation between hpo-miR396b and GRF was validated under different stresses. Figure 8A is provided to present spatio-temporal expression of miR396b in pitaya.

Point 12: Please edit your manuscript more carefully: analysis. Based, SorgHpm bicolor.

Response 12: We made the corresponding revision in the new version.

Point 13: Please explain why did you use term crop for pitaya.

Response 13: We replaced “crop” with “fruit” in Abstract in the new version.

Reviewer 2 Report

Dear Authors,

Reviewer comments ijms-482136

The manuscript entitled „Conserved miR396b targeting GRF genes is involved in the abiotic stress response of pitaya (Hylocereus polyrhizus)“ represents a useful study aimed at an investigation of target genes of microRNA396 which is known to be responsive to various abiotic stress factors including osmotic stress (PEG), high and low temperatures, salt (NaCl) and ABA treatments. The study is focused on an investigation of transcript levels of hpo-miR396b and its target gene HpGRF6 in pitaya (Hylocereus polyrhizus) but also an overview of other hpo-miR396b target genes is provided in Table A1.

The presented results are surely worth publishing. I have only a few formal comments on the manuscript.

Abbreviations have to be listed in an alphabetical order in Abbreviations list.

Figure 8 legend: The kind of statistical test used for determination of significant differences between the transcripts has to be given in the figur legend. Based on the information in Statistical analysis section, the authors used Duncan´s multiple range test following ANOVA analysis??

In Materials and methods, part 4.1. Plant material and stress treatment, the pitaya genotype used in the experiments as well as the source )institution) from which the plant material was obtained have to be given.

Materials and methods, part 4.7. Statistical analysis: The first sentence starting with „The relative expression level was calůculated with the formula…“ has to be replaced to the preceding section 4.6. qRT-PCR analysis.

Startistical analysis: The authors wrote only about Duncan´s multiple range test but it is usual that this test follows ANOVA analysis. Thus I think that the sentence has to be modified as follows: „All experiments were repeated three times, and all data were analyzed by two-way ANOVA?? followed by Duncan´s multiple range test.“

Further formal comments:

Materials and methods: Use SI units only in the Scientific text, i.e., use „mm3“ instead of „μL“ for volume units.

Formal comments on the text:

Abstract, line 8: Add a comma both before and after the word „generally“ in the sentence „…and, generally, a more recent evolutionary relationship in the intra-family has been demonstrated.“

Abstract, line 19: Use SI units for volume, i.e., write „100 mg dm-3“ (NOT „100 mg L-1“).

Introduction, page 2, line 5: Correct the words „for understanding their function.“ (NOT „for understanding theses function.“) in the sentence „Thus, elucidation of the expression patterns of miRNAs to their corresponding targets at specific growth stages or in certain growth environments is of great importance for understanding their function.“

Introduction, page 2, line 27: Add the word ůinů in the sentence „Unfortunately, there are some limitations in its yield potential,…“

Introduction, page 2, line 31: Replace the words „were determiend“ by the words „were applied“ in the sentence „Previously, small RNA-seq, RNA-seq, and degradome-seq were applied in pitaya under exposure to drought stress,…“

Results, Page 3, line 6: Add the term „minimum free energy“ to the sentence „…however, their loops varied in sequence and minimum free energy“.

Results, page 6, line 3: Replace the word „conversation“ by the word „conservation“ in the sentence „…indicating high conservation of the three GRFs.“ (or „indicating high conservation among the three GRFs.“)

Discussion, page 8, line 45: Add the word „to“ in the sentence „Considering all evidence, it is clear that miR396-targeted GRFs play important roles in the response to biotic and abiotic stresses,..“ (“in the response to biotic and abiotic stresses,…“).

Final recommendation: Accept after a minor revision.

Author Response

Point 1: Abbreviations have to be listed in an alphabetical order in Abbreviations list.

Response 1: The abbreviations are listed in an alphabetical order in Abbreviations list.

Point 2: Figure 8 legend: The kind of statistical test used for determination of significant differences between the transcripts has to be given in the figure legend. Based on the information in Statistical analysis section, the authors used Duncan´s multiple range test following ANOVA analysis?

Response 2: “All data were analyzed by two-way ANOVA, and followed by Duncan’s multiple range test. were added in Figure 8 legend in the new version. The methods of Duncan´s multiple range test following ANOVA analysis were used in the statistical analysis section.

Point 3: In Materials and methods, part 4.1. Plant material and stress treatment, the pitaya genotype used in the experiments as well as the source (institution) from which the plant material was obtained have to be given.

Response 3: Two sentences, e.g. “Given that the stem is the primary organ of pitaya seedlings, it was used to examine genes expression changes in response to abiotic stresses. The stem of pitaya (H. polyrhizus) was obtained from research greenhouse, Ministry of Education, Institute of Agro-bioengineering, Guizhou University.” and “ which were provided by the Institute of Fruit Trees, Guizhou Province Academy of Agricultural Sciences” were added in lines 457-460 and 472-473 of the revised version, respectively.

Point 4: Materials and methods, part 4.7. Statistical analysis: The first sentence starting with The relative expression level was calůculated with the formula…has to be replaced to the preceding section 4.6. qRT-PCR analysis.

Response 4: We made the corresponding revision in line 486 of the revised version.

Point 5: Startistical analysis: The authors wrote only about Duncan´s multiple range test but it is usual that this test follows ANOVA analysis. Thus I think that the sentence has to be modified as follows: All experiments were repeated three times, and all data were analyzed by two-way ANOVA?? followed by Duncan´s multiple range test.

Response 5: following two-way ANOVA analysis were added in line 546 of the revised version.

Point 6: Materials and methods: Use SI units only in the Scientific text, i.e., use mm3instead of μL for volume units.

Response 6: Thank you for your suggestions. It is true that the volume units is “mm3” in SI units, however, the instructions of the relevant kits use μL for volume units, and it is common to use “μL in the relevant molecular biology research. In our opinions, it is more appropriate to use “μL for volume units.

Point 7: Abstract, line 8: Add a comma both before and after the word “generally” in the sentence “…and, generally, a more recent evolutionary relationship in the intra-family has been demonstrated.”

Response 7: A comma was inserted after the word ‘generally’ in line 18 of the revision.

Point 8: Abstract, line 19: Use SI units for volume, i.e., write “100 mg dm-3” (NOT “100 mg L-1”).

Response 8: Thank you for your recommendation, the response is the same as Response 6.

Point 9: Introduction, page 2, line 5: Correct the words for understanding their function. (NOT for understanding theses function.) in the sentence Thus, elucidation of the expression patterns of miRNAs to their corresponding targets at specific growth stages or in certain growth environments is of great importance for understanding their function.

Response 9: The word “theses” was corrected for “their” in line 46 of the revised version.

Point 10: Introduction, page 2, line 27: Add the word ůinů in the sentence Unfortunately, there are some limitations in its yield potential,…”.

Response 10: The word “in” was added in line 69 of the new version.

Point 11: Introduction, page 2, line 31: Replace the words “were determiend” by the words “were applied” in the sentence “Previously, small RNA-seq, RNA-seq, and degradome-seq were applied in pitaya under exposure to drought stress,…”.

Response 11: We replaced “were determiend” with “were applied” in line 73 of the new version.

Point 12: Results, Page 3, line 6: Add the term minimum free energy to the sentence …however, their loops varied in sequence and minimum free energy.

Response 12: “ and minimum free energy” was added in line 98 of the new version.

Point 13: Results, page 6, line 3: Replace the word conversation by the word conservation in the sentence …indicating high conservation of the three GRFs. (or indicating high conservation among the three GRFs.)

Response 13: The Original sentence was replaced by “indicating high conservation among the three GRFs.” in line 168-169 of the new version.

Point 14: Discussion, page 8, line 45: Add the word “to” in the sentence “Considering all evidence, it is clear that miR396-targeted GRFs play important roles in the response to biotic and abiotic stresses,..” (“in the response to biotic and abiotic stresses,…”).

Response 14: The word “to” was added in line 388 of the new version.

Round  2

Reviewer 1 Report

The manuscript is almost ready but the authors should improve the visibility of chloroplasts on Figure 7 panel.

Author Response

Comments and Suggestions for Authors

The manuscript is almost ready but the authors should improve the visibility of chloroplasts on Figure 7 panel.

Response : Thank you for your suggestions. We have improved the visibility of chloroplasts on Figure 7 panel in the new version.